# Magnetically Activated Piezoelectric 3D Platform Based on Poly(Vinylidene) Fluoride Microspheres for Osteogenic Differentiation of Mesenchymal Stem Cells

**DOI:** 10.3390/gels8100680

**Published:** 2022-10-20

**Authors:** Maria Guillot-Ferriols, María Inmaculada García-Briega, Laia Tolosa, Carlos M. Costa, Senentxu Lanceros-Méndez, José Luis Gómez Ribelles, Gloria Gallego Ferrer

**Affiliations:** 1Centre for Biomaterials and Tissue Engineering (CBIT), Universitat Politècnica de València, 46022 Valencia, Spain; 2Biomedical Research Networking Center on Bioengineering, Biomaterials and Nanomedicine, Carlos III Health Institute (CIBER-BBN, ISCIII), 46022 Valencia, Spain; 3Experimental Hepatology Unit, Health Research Institute La Fe (IIS La Fe), 46026 Valencia, Spain; 4Physics Centre of Minho and Porto Universities (CF-UM-UP), University of Minho, 4710-057 Braga, Portugal; 5Laboratory of Physics for Materials and Emergent Technologies, LapMET, University of Minho, 4710-057 Braga, Portugal; 6Institute of Science and Innovation for Bio-Sustainability (IB-S), University of Minho, 4710-057 Braga, Portugal; 7BCMaterials, Basque Centre for Materials, Applications and Nanostructures, UPV/EHU Science Park, 48940 Leioa, Spain; 8IKERBASQUE, Basque Foundation for Science, 48009 Bilbao, Spain

**Keywords:** mesenchymal stem cells, osteoblastogenesis, piezoelectricity, poly(vinylidene) fluoride, hydrogel

## Abstract

Mesenchymal stem cells (MSCs) osteogenic commitment before injection enhances bone regeneration therapy results. Piezoelectric stimulation may be an effective cue to promote MSCs pre-differentiation, and poly(vinylidene) fluoride (PVDF) cell culture supports, when combined with CoFe_2_O_4_ (CFO), offer a wireless in vitro stimulation strategy. Under an external magnetic field, CFO shift and magnetostriction deform the polymer matrix varying the polymer surface charge due to the piezoelectric effect. To test the effect of piezoelectric stimulation on MSCs, our approach is based on a gelatin hydrogel with embedded MSCs and PVDF-CFO electroactive microspheres. Microspheres were produced by electrospray technique, favouring CFO incorporation, crystallisation in β-phase (85%) and a crystallinity degree of around 55%. The absence of cytotoxicity of the 3D construct was confirmed 24 h after cell encapsulation. Cells were viable, evenly distributed in the hydrogel matrix and surrounded by microspheres, allowing local stimulation. Hydrogels were stimulated using a magnetic bioreactor, and no significant changes were observed in MSCs proliferation in the short or long term. Nevertheless, piezoelectric stimulation upregulated RUNX2 expression after 7 days, indicating the activation of the osteogenic differentiation pathway. These results open the door for optimising a stimulation protocol allowing the application of the magnetically activated 3D electroactive cell culture support for MSCs pre-differentiation before transplantation.

## 1. Introduction

Mesenchymal stem cells (MSCs) promote the functional repair of bone injuries due to their osteogenic differentiation potential [1]. Osteogenic commitment before transplantation shows better results in regeneration therapies by enhancing mineral deposition and integration in the damaged site compared to undifferentiated MSCs injection [2,3,4]. The standard pre-differentiation protocol employs osteoinductive cell culture media containing dexamethasone, β-glycerophosphate and ascorbic acid in a tissue culture plate. Dexamethasone lacks specificity and produces mixed populations containing fat cells [5]. Moreover, MSCs dedifferentiate once the biochemical stimulus is removed unless deployed with an extracellular matrix (ECM) resembling the environment [6]. As an alternative, physical cues are suitable and specific candidates for MSCs differentiation control [7,8,9,10,11], solving the off-target problem of dexamethasone supplementation [12].

MSCs’ piezoelectric stimulation has gained attention since the discovery of bone’s inherent piezoelectricity [13]. MSCs differentiation in vivo occurs in an electrically active environment due to collagen type I fibres conforming to 90% of bone’s organic ECM [14].

-NH- and -CO- groups in the protein amino acids generate a permanent polarisation in the fibres. When mechanical stress is applied, the piezoelectric effect changes the surface charge [15]. This bioelectricity is associated with bone’s ability to grow and remodel [16,17], so it may induce MSCs differentiation towards the osteogenic lineage.

Poly(vinylidene) fluoride (PVDF) and its high piezoelectric coefficient, when crystallised in β-phase, has been widely used in the tissue engineering field for piezoelectric stimulation [15]. Different morphologies, including films, porous membranes, fibres, or microspheres, have been developed [18], offering an electroactive milieu for MSCs’ growth and differentiation [19,20,21,22,23,24]. PVDF coupled with magnetostrictive nanoparticles, such as cobalt ferrite oxide (CoFe_2_O_4_; CFO), provides piezoelectric stimulation at the cell culture level using a magnetic bioreactor [22,25,26]. Their combination generates a magnetoelectric composite material. When exposed to an external magnetic field, the magnetostrictive phase is deformed, deforming the polymeric matrix and resulting in a dielectric polarisation variation ascribable to the piezoelectric properties of the polymer [27,28]. This approach suits translational applications based on the wireless nature of the magnetic field, allowing minimally invasive stimulation strategies in vitro. PVDF-CFO composites have been produced by different methods in different shapes, including films [29,30], membranes [31], fibres [26], spheres [25,32] or scaffolds [22]. Nevertheless, the use of this strategy to stimulate MSCs for pre-differentiation approaches is scarcely reported, especially using 3D cell culture supports.

We hypothesise that piezoelectric stimulation may induce specific MSCs osteogenic commitment for pre-differentiation approaches by using an electroactive and biomimetic cell culture platform. To prove our hypothesis, we have designed a 3D platform based on a gelatin hydrogel containing PVDF-CFO electroactive microspheres together with MSCs, stimulable using a magnetic bioreactor. To do so, PVDF microspheres with and without CFO have been produced, characterised, and encapsulated in the hydrogel. Effects of piezoelectric stimulation on MSCs proliferation and osteogenic differentiation have been tested by metabolic activity, gene expression and alkaline phosphatase production. The study of gene expression and an early osteogenic marker, such as alkaline phosphatase, at 7 and 14 days allows a quick evaluation of cell commitment. They provide a starting point to adjust several variables that may affect the stimulation outcome (cell culture media, stimulation parameters and times).

As far as the authors know, this is the first time that this kind of platform has been described to study the effect of piezoelectric stimulation on MSCs’ osteogenic pre-differentiation *in vitro* before cell transplantation. We have generated an electroactive and biomimetic environment that recapitulates several aspects of the bone niche. Gelatin hydrogels are easily processable for cell recovery after stimulation, obtaining a population of committed MSCs for regeneration therapies alone or in combination with a biodegradable scaffold.

## 2. Results and Discussion

### 2.1. Microsphere Characterisation

PVDF and PVDF-CFO microspheres were produced by electrospray. Electrospray is a one-step technique allowing to produce narrow size distributions of microspheres, overcoming the limitations of emulsion-based approaches. As presented in Figure 1a, the microspheres show a diameter distribution of 0.5 to 6 µm in the case of PVDF and 0.5 to 4 µm for PVDF-CFO, with mean diameters of 2.46 ± 1.08 and 1.64 ± 0.6 µm, respectively. 

Adding CFO nanoparticles to the PVDF solution increased its electrical properties and viscosity, hindering droplet formation [33]. A lower PVDF concentration was used, originating differences in size distribution and mean diameters. PVDF concentrations from 4 to 10% (*w*/*v*) lead to microsphere production, with increasing diameters, whereas higher concentrations favour fibre formation, as described by Correia et al. [34]. This concentration range allows a semi-dilute moderate entanglement of the polymer chains, giving rise to round dense microspheres, as presented in Figure 1a.

CFO incorporation and the amount were assessed by a vibrational sample magnetometer (VSM). The typical hysteresis loop for PVDF-CFO nanocomposites is presented in Figure 1c. The saturation magnetisation of the analysed sample was compared with pure CFO powder by applying Equation (1) (see Materials and Methods section), revealing a real CFO content of 7.8 ± 1.8% (*w*/*w*). The composite solution concentration was 20% (*w*/*w*), which indicates a CFO loss of more than 50% during microsphere production. These results agree with the ones obtained by Gonçalves et al. [25], where different concentrations of CFO in the composite solution were compared to the final concentration present in the electrosprayed spheres. It revealed that the content in the multiferroic spheres is always lower than in the solution due to the higher density of the CFO that causes the settling of the nanoparticles on the bottom of the syringe during the manufacturing process.

CFO incorporation was observed by cross-sectioning the microspheres. Figure 1b shows a representative cross-section image of PVDF-CFO microspheres, where white arrows point to CFO nanoparticles. CFO aggregates are in direct contact with the polymer matrix. Under the applied magnetic field, CFO nanoparticles exert a combination of mechanical and magnetostrictive actions on the piezoelectric phase favouring the magnetoelectric effect on the surrounding medium as already described by Gonçalves et al. for the same type of microspheres [25].

PVDF can present five polymorphs (α, β, γ, δ and ε), but not all of them are electroactive. α β, and γ are the most commonly obtained phases by standard manufacturing techniques. Piezoelectric stimulation of different cell types requires the presence of an electrically active phase, and β-phase crystallisation is usually preferred due to its high piezoelectric coefficient. PVDF polymorphic polymers’ vibrational spectra via FTIR have been validated for phase identification [35]. The method is based on identifying absorption peaks that exclusively appear in one of the phases. In the case of the α-phase, the non-electroactive one, its characteristic peaks are around 410, 489, 532, 614, 762, 795, 854, 975, 1149, 1209, 1383 and 1423 cm^−1^, being 762 cm^−1^ the one used to unequivocally identify it [35,36]. In the case of the electrically active phases β and γ, their identification using FTIR has been a matter of debate in recent years. Traditionally, the peak around 840 cm^−1^, present in the samples (Figure 1d), has been considered a characteristic β-phase peak. Nevertheless, both polymorphs share this band, but it appears as a strong one only for the β-phase. For the γ-phase, it is shown as a shoulder of the 833 cm^−1^ peak [37,38]. The bands at 1279 and 1234 cm^−1^ are exclusive of β and γ phases, respectively, and are consistently used to identify them [35].

The representative FTIR-ATR spectra of both microspheres are displayed in Figure 1d, showing α-phase characteristic bands at 489, 762 and 975 cm^−1^, highlighted in the graph. The presence of the 1279 cm^−1^ band corroborates the existence of the β-phase in the microspheres. The strong band at 840 cm^−1^ is used to quantify the percentage of the crystalline phase in the samples applying Equation (2). Quantification revealed that the percentages of β-phase in PVDF and PVDF-CFO microspheres were 84.8 ± 2.9% and 84.6 ± 3.3%. Electrospray favours β-phase crystallisation because it occurs at temperatures lower than 70 °C [39,40]. The high voltage applied to the initial solution, the high stretching ratio of the jet [41], and the incorporation of fillers in the PVDF matrix [29] also improve the β-phase content. Nevertheless, their contribution is negligible compared to the effect of the solvent evaporation temperature [18].

Finally, the thermal properties of the electrosprayed microspheres were investigated using differential scanning calorimetry (DSC). Since PVDF is a semi-crystalline polymer, its crystalline regions are immersed in an amorphous polymer matrix, and the degree of crystallinity (*X_c_*) can be calculated from the obtained melting enthalpies applying Equation (3). Figure 1e shows the presence of endothermic peaks around 170 °C. Melting temperatures (T_m_) corresponding to PVDF, and PVDF-CFO microspheres were similar, being 167.9 and 168.8 °C, respectively. The main difference between both thermograms was the presence of a double endothermic peak in the samples containing magnetostrictive nanoparticles. This double peak can be attributed to crystalline imperfection since the presence of CFO can generate crystal defects in the sample. These data correlate with the *X_c_*, being higher in PVDF microspheres (58%) compared to PVDF-CFO (54%). Martins et al. [29] described that composite materials tend to have a lower degree of crystallinity than pristine PVDF due to the presence of CFO.

### 2.2. MSCs Viability and Distribution within the 3D Construct

PVDF or PVDF-CFO microspheres and MSCs were encapsulated in gelatin hydrogels to generate an electroactive 3D cell culture platform. An illustration of the system is presented in Figure 1.

MSCs viability was evaluated after 24 h of encapsulation since CFO has proven cytotoxic for this cell type [42]. Figure 2a shows representative images of the hydrogels, where MSCs nuclei were stained with Hoechst and dead cells nuclei with propidium iodide. Gel-PVDF and Gel-PVDF-CFO hydrogels images show the presence of the microspheres, which appear as black dots in the case of the ones containing ferrite. After quantification (Figure 2b), no significant differences were observed compared to the gelatin hydrogels without microspheres, used as viability control [43]. These results revealed that the magnetostrictive nanoparticles were enclosed inside the polymer matrix, reducing their cytotoxicity, and not affecting MSCs viability. On the other hand, including electroactive microspheres inside the gelatin hydrogel successfully resulted in a viable 3D cell culture platform that could stimulate MSCs.

This is not the first time a hydrogel-based magnetoelectric microenvironment has been described for cell stimulation. Hermenegildo et al. [44] used methacrylated Gellan Gum to encapsulate PVDF-CFO microspheres. Nevertheless, cells were not embedded in the hydrogels but were seeded on the surface, reducing the effect of the local piezoelectric stimulation. Similarly, Carvalho et al. [45] used the same hydrogel but combined it with poly-L-lactic acid (PLLA)-CFO microspheres. In these cases, cells were both seeded on the surface and injected inside the hydrogels to generate a 3D environment. MSCs osteogenic differentiation induced by piezoelectric stimulation was not tested in any of these works, although similar platforms were described.

After assessing MSCs viability, a closer look into cell spreading and distribution within the hydrogel was taken by cryo-sectioning the samples after 1 and 14 days of culture. Figure 3 shows an even cell distribution along the gelatin matrix in all samples. In the case of the hydrogels containing microspheres, these appear uniformly distributed, surrounding the cells. Regarding cell spreading, after 24 h, cells show a rounded morphology with a scarcely developed cytoskeleton.

After 14 days in culture, MSCs present a fibroblastic morphology with a spindled shape, characteristic of this cell type in adherent substrates. Gelatin is a molecular derivative of type I collagen; although less organised, it is biocompatible, cheaper and preserves the linear tripeptide Arginine-Glycine-Aspartate (RGD) recognition sequence that binds to several integrin proteins promoting cell attachment, migration, and survival [46], as demonstrated by the images taken after 14 days. MSCs encapsulation in a gelatin hydrogel provides the active biological cues lacking in PVDF chemical structure, avoiding PVDF surface modification in a 3D environment and allowing ECM-cell and cell-cell interaction.

These results are supported by Appendix A and 1, where a 3D reconstruction of a non-cryo-sectioned Gel-PVDF hydrogel is shown (see video and Z projections in supplementary materials). In their interior, MSCs are completely elongated and form a 3D interconnected network.

### 2.3. Effect of Piezoelectric Stimulation on MSCs Proliferation and Osteogenic Differentiation

MSCs proliferation was determined after 2, 7, 14 and 21 days in non-stimulated (NS) and stimulated (S) conditions. Gelatin hydrogels without microspheres were used as controls to evaluate the effect of the magnetic field generated by the bioreactor. For each condition tested, Gel, Gel-PVDF and Gel-PVDF-CFO were compared with their stimulated counterpart and the rest of the conditions at every time point. Gel-PVDF-CFO stimulated condition is the only one able to provide piezoelectric stimulation due to the presence of the magnetostrictive nanoparticles and the magnetic field.

As presented in Figure 4a, no significant differences in proliferation are observed after 2, 7 and 14 days. A significant change in proliferation can be noted after 21 days between Gel NS, and Gel-PVDF-CFO S. The only significant difference in proliferation appears between these two conditions, and there is no change compared with the non-stimulated control PVDF-CFO NS. This fact may be motivated by the apparent reduction in absorbance observed in Gel NS between 14 and 21 days. Cells may be scaping the hydrogels in this type of sample, but further tests will be needed to extract conclusions from this fact.

Nevertheless, no changes in proliferation are observed when comparing Gel-PVDF-CFO stimulated and non-stimulated at any of the studied time points. The magnetic field generated by the bioreactor does not influence MSCs proliferation since there is no difference between Gel NS and Gel S hydrogels. In the same way, the lack of significant difference in Gel-PVDF-CFO NS and Gel-PVDF-CFO S demonstrates that piezoelectric stimulation has no positive or negative influence on MSCs proliferation using this 3D cell culture support with the applied stimulation parameters. These results differ from those obtained by Fernandes et al. [22], where an increase in cell proliferation was observed after 4 days when comparing static and dynamic conditions. Although they used a 3D scaffold and the stimulation parameters applied were the same, the study was performed using MC3T3-E1 pre-osteoblast cells. This different cell type might respond in a different way to piezoelectric stimulation. Carvalho et al. [45] also described an increase in proliferation after 3 days when using a combination of a gelatin hydrogel with embedded PLLA-CFO microspheres stimulated with similar parameters, compared with non-stimulated samples. Even though the cell culture platform is similar to the one described here, they also used the MC3T3-E1 cell line, which, as already mentioned, can respond differently to this kind of stimulation.

Proliferation does not change over time in hydrogels with the same composition. This phenomenon was also reported by Moulisová et al. [47], showing no MSCs proliferation in gelatin hydrogels when using a basal medium after 14 days. In our case, the presence of microspheres alone or its combination with a magnetic field in the 3D support does not alter MSCs proliferation behaviour in the gelatin matrix.

The effect of electromechanical stimulation on MSCs osteogenic differentiation was tested after 7 and 14 days in culture by analysing gene expression of characteristic osteogenic markers and alkaline phosphatase (ALP) activity. For differentiation assays, piezoelectric stimulation was combined with a commercial osteogenic medium.

Figure 4b shows the relative expression of ALP, collagen type I (COL I) and runt-related transcription factor 2 (RUNX2) at 7 days of culture, and ALP, COL I and osteocalcin (OCN) at 14 days. RUNX2 expression, an early osteogenic marker, increases in Gel-PVDF-CFO stimulated samples compared to non-stimulated after 7 days. Nevertheless, no differences are observed in mid-stage markers ALP and COL I. After 14 days, COL I expression is enhanced in Gel-PVDF S, where no piezoelectric stimulation is experienced since it does not contain magnetostrictive nanoparticles.

Gene expression agrees with ALP activity measured at protein level after 7 days of culture, which shows no differences between conditions (Figure 4c). However, after 14 days, ALP activity increases in the non-stimulated condition compared to stimulated Gel-PVDF and Gel-PVDF-CFO. It contradicts gene expression results at this time point. ALP expression peak is usually reported after 10 days of culture. The gene expression analysis at only two-time points may have missed it, resulting in differences only shown at the protein level.

RUNX2 is a key transcriptional regulator of osteoblast differentiation and bone formation. Its activation by phosphorylation in MSCs leads to osteogenic commitment and the subsequent expression of downstream genes involved in the differentiation process, such as alkaline phosphatase, collagen type I, osteopontin and osteocalcin [48]. Piezoelectric stimulation in the proposed 3D cell culture platform can activate the osteogenic differentiation pathway with an increased expression of the master regulator RUNX2; nevertheless, this effect is not sustained in time. The following expression of mid-stage markers, such as ALP and COL I, is neither enhanced nor ALP activity.

Several factors may influence the effect of piezoelectric stimulation on MSCs differentiation towards the osteogenic lineage [49]. The selection of cell culture media, stimulation parameters and treatment times are non-trivial choices when designing stimulation experiments. The lack of a standardised stimulation protocol for MSCs makes comparing published results difficult.

Regarding stimulation parameters, the use of different bioreactors for activating piezoelectric substrates comprises the use of different stimulation parameters in literature. Treatment times are also a matter of debate when stimulating MSCs for differentiation. In this work, a stimulation program divided into an active period of 16 h based on 5 min of magnetic stimulation and 25 min of resting time, followed by a non-active period of 8 h, was selected to simulate daily human activity. Stimulation was applied for the total duration of the culture. Again, despite the influence that treatment time may have in a highly orchestrated and time-dependent process such as osteoblastogenesis, no studies have been published on its optimisation when using piezoelectric cell culture supports. Other types of electrical stimulation, the one using conductive cell culture supports and an external power generator, have explored the effectiveness of diverse factors regarding treatment time. First, stimulation time per day, thus, the number of hours that cells are subjected to stimulation each culture day; second, the number of days those cells receive the stimulation along the duration of the culture and last, the moment where the stimulation is introduced (early, mid or late stages).

Wechsler et al. [50] demonstrated that for MSCs cultured in indium tin oxide-coated glass, the optimal stimulation time per day was 6 h rather than shorter (1–3 h) and longer (24 h). On another note, Zhu et al. [51] artificially divided the 21-day culture time into 7-day periods and applied ES for 1.5 h a day for the selected period. Day 1 to 7 stimulation improved the expression of MSCs bone-related markers rather than an application from day 8 to 14 and 15 to 21. This corroborates the results obtained by Hu et al. [52], where stimulation was applied for 4 h on a selected day (days 0, 2, 4, 6, 8, 10 and 12), demonstrating that day 8 was the optimal one since MSCs showed a higher level of mineral deposition after 14 days, supported by the upregulation of osteogenic genes.

These data reveal that stimulation application for osteogenic differentiation induction is a time-dependent process, and if optimised, it should only be applied at specific time points. In this case, application along the culture may have a detrimental effect on the differentiation process, as shown by the results in gene expression and ALP activity after 14 days.

Finally, media selection in combination with electromechanical stimulation may influence the result. In this work, a commercial osteogenic medium was used. Nevertheless, MSCs’ osteogenic fate determination has been widely reported as a result of using piezoelectric biomaterials as cell culture supports in the presence of a growth medium [23,24,53,54,55,56,57]. The strong effect of biochemical inducers present in the osteogenic medium may have covered the impact of piezoelectric stimulation. Even so, this physical cue, combined with the osteogenic medium, can trigger RUNX2 expression to a greater extent than osteogenic-induced non-stimulated samples.

These results leave room for improvement and open the door for future optimisation of the stimulation protocol regarding treatment times and parameters. Studying different cell culture media may also be beneficial for assuring a stable osteogenic phenotype before MSCs administration. The next steps will also include cell recovery from the hydrogel after stimulation and their injection alone or in combination with a biodegradable scaffold.

## 3. Conclusions

PVDF and PVDF-CFO microspheres were produced by electrospray, a reliable technique that allows PVDF crystallisation in β-phase, its most electroactive polymorph. The microspheres incorporated CFO nanoparticles in their interior without compromising their crystallinity degree, similar to the one obtained in non-containing magnetostrictive nanoparticles. Microspheres, together with MSCs, were successfully encapsulated in a tyraminated gelatin hydrogel, generating a 3D cell culture platform. This platform resulted in non-cytotoxic for MSCs, meaning that the CFO was well enclosed in the microsphere polymer matrix. MSCs and microspheres were evenly distributed in the gelatin matrix after 1 day of culture, allowing local stimulation. After 14 days, MSCs showed a well-developed cytoskeleton with a fibroblastic-like shape, forming an interconnected 3D network cultured in an expansion medium without stimulation. Magnetically induced piezoelectric stimulation had no positive or negative influence on MSCs proliferation in the proposed 3D cell culture platform. Regarding MSCs’ osteogenic differentiation, combination with osteogenic medium revealed an increase of RUNX2 expression after 7 days compared to non-stimulated samples, indicating a stronger activation of the osteogenic differentiation pathway.

## 4. Materials and Methods

### 4.1. Microsphere Production by Electrospray Technique

PVDF microspheres with and without magnetostrictive nanoparticles were obtained by electrospray technique, adapting the protocol from references [25,34]. A 9% (*w*/*v*) PVDF (Solef^®^ 6010 PVDF Homopolymer, Solvay, Brussels, Belgium) solution was prepared by dissolving the polymer in a mixture 85/15 (*v*/*v*) of N,N-dimethyl formamide ((DMF) synthesis grade, Scharlab, Barcelona, Spain) and tetrahydrofuran ((THF) synthesis grade, Scharlab) at room temperature for 2 h. The composite solution was prepared by dispersing Cobalt Ferrite Oxide (CoFe_2_O_4_; CFO) nanoparticles (Nanoamor, 35–55 nm diameter) at a concentration of 20% (*w*/*w*) in DMF solvent containing 1% (*v*/*v*) Triton X-100 (Sigma-Aldrich, St. Louis, MI, USA) to prevent particle agglomeration. A high-performance dispersing machine (ULTRA-TURRAX ^®^, IKA, Staufen, Germany) at 6500 rpm was used to disperse the CFO for 30 min, and after that, PVDF (4% (*w*/*v*)) and THF solvent were added. PVDF concentration was reduced for PVDF-CFO microsphere manufacturing due to the presence of the MNPs in the solution, which produced an increase in viscosity and the dielectric constant. The mixture was stirred for another hour until the complete dissolution of the polymer.

The solutions were placed in a commercial plastic syringe fitted with a steel needle of 1.7 mm inner diameter. Electrospray was conducted by applying a voltage of 20 kV with a high-voltage power supply (Glassman High Voltage, Inc., High Bridge, NJ, USA). A syringe pump (SyringePump) pumped the solution at a rate of 2 mL/h. Microspheres were collected in a grounded conductive aluminium collector immersed in a bath of liquid nitrogen [58] placed at 20 cm from the needle tip. The syringe’s content was replaced every 20 min for the composite solution to avoid nanoparticle precipitation.

Microspheres were rinsed with ethanol, sonicated in an ultrasound bath, and sieved with a 40 μm strainer to eliminate polymer aggregates.

### 4.2. Microsphere Characterisation

#### 4.2.1. Field Emission Scanning Electron Microscopy

Microspheres were morphologically characterised using a field emission scanning electron microscopy (FESEM) (AURIGA compact, Zeiss, Jena, Germany) with an accelerating voltage of 2 kV. Samples were coated with platinum following a standard sputtering protocol for 90 s (JFC 1100, JEOL, Tokyo, Japan). For observation of nanoparticle distribution, PVDF-CFO microspheres were cross-sectioned using a focused ion beam (FIB) device coupled to FESEM, and images were taken after sectioning. Microsphere diameter was assessed from FESEM images. At least 700 microspheres from three independent batches were measured using ImageJ software (National Institutes of Health, Bethesda, MD, USA).

#### 4.2.2. Vibrational Sample Magnetometer

Magnetic properties and nanoparticle content in the PVDF-CFO microspheres were determined using a Microsense 2 Tesla vibrational sample magnetometer (VSM). Magnetisation loops M(H) were evaluated up to ± 18 kOe, and pure CFO saturation magnetisation value (60 emu/g) was compared to the one obtained in the composite samples to obtain the effective filler content in the microspheres by means of Equation (1) [25]:(1)$$CFO\;wt\%=\frac{Saturation\;magnetization\;microspheres}{Saturation\;magnetization\;pure\;CFO}\times 100$$

Measurements were taken from samples produced in three different batches.

#### 4.2.3. Fourier Transform Infrared Spectroscopy

Fourier transform infrared spectroscopy (FTIR) has proven to be an effective technique for determining the electroactive phase content of PVDF. Gregorio and Cestari [39] described a method based on identifying the characteristic peaks at 840 cm^−1^ and 762 cm^−1^, which are associated with the presence of β and α phases, respectively. They can be quantified using Equation (2):(2)Fβ=AβKβKα  Aα +Aβ 

The assumption of Lambert-Beer’s law is a requirement to apply Equation (2). K_α_ and K_β_ values are 6.1 × 10^4^ and 7.7 × 10^4^ cm^2^/mol, respectively, and correspond to the characteristic absorption coefficients at 762 and 840 cm^-1^ obtained from pristine α or β-phase samples [39]. A_α_ and A_β_ were obtained using a spectrometer in ATR mode (ALPHA FTIR, Bruker) in the wavenumber range of 4000 to 400 cm^-1^, at a resolution of 4 cm^-1^.

Measurements were taken from samples produced in three different batches.

#### 4.2.4. Differential Scanning Calorimetry

PVDF’s semicrystalline nature requires its thermal characterisation to determine the crystallinity degree (*X_c_*). Samples were evaluated by differential scanning calorimetry (DSC) in a DSC 8000 (PerkinElmer). A mass of 2–4 mg of microspheres was encapsulated in aluminium pans and heated from 0 °C to 200 °C at a heating rate of 20 °C/min in a dry nitrogen atmosphere.

The degree of crystallinity was calculated with the obtained data by applying Equation (3) [29]: (3)$$Xc=\frac{\Delta H_{m}}{w_{PVDF\;}\left(x\Delta H_{\alpha }+y\Delta H_{\beta }\right)}$$

∆H_m_ is the melting enthalpy of PVDF and PVDF-CFO microspheres measured by DSC. ΔH_α_ and ΔH_β_ values are 93.07 J/g and 103.4 J/g, respectively, and correspond to the melting enthalpies of α and β phases total crystalline samples [59]. The magnetic properties allow obtaining the percentage of polymer present in the microspheres (mass fraction; w_PVDF_) and FTIR measurements provide the percentage of α and β phases (x and y). 

### 4.3. Microsphere Polarisation

Microspheres were polarised by the corona poling method in a homemade poling chamber for 60 min at 100 °C and ~10 kV to maximise their macroscopic piezoelectric response.

### 4.4. Cell Response

#### 4.4.1. Microsphere and MSCs Encapsulation in 3D Gelatin Hydrogels

Human bone marrow mesenchymal stem cells (PromoCell, Heidelberg, Germany) together with PVDF (Gel-PVDF) or PVDF-CFO (Gel-PVDF-CFO) microspheres were encapsulated in gelatin hydrogels to generate a 3D cell culture platform. An illustration of the 3D cell culture platform and its components are presented in Figure 1.

Gelatin (from porcine skin, gel strength 300, type A, Sigma-Aldrich) was conjugated with tyramine (Sigma-Aldrich) following the protocol described in reference [47], based on N-hydroxysuccinimide (NHS) (Sigma-Aldrich) and 1-ethyl-3-(3 dimethylaminopropyl) carbodiimide hydrochloride (EDC) (Iris Biotech GmbH, Marktredwitz, Germany) chemistry. Tyramine conjugation allows gelatin enzymatic in situ cross-linking, as described in reference [60].

To obtain the hydrogels, tyramine conjugated gelatin was dissolved at 2% (*w*/*v*) in Calcium-free Krebs Ringer Buffer (CF-KRB; 115 mM sodium chloride, 5 mM potassium chloride, 1 mM potassium dihydrogen phosphate, and 25 mM 4-(2-hydroxyethyl)piperazine-1-ethanesulphonic acid)) for 30 min at 37 °C. Hydrogels were prepared with 80% (*v/v*) of the gelatin solution, 10% (*v*/*v*) horseradish peroxidase ((HRP) Sigma-Aldrich) at 12.5 U/mL (1.25 U/mL in the final volume), and 10% (*v*/*v*) H_2_O_2_ (Sigma-Aldrich) 20 mM (2 mM in the final volume). All solutions were sterile filtered after complete dissolution. 50 µL hydrogels were used for cell culture assays. For hydrogels containing microspheres, those were added at 0.6% (*w*/*v*) concentration. Microspheres were weighted, resuspended in ethanol and placed in an ultrasound bath to avoid agglomeration. Microspheres were sterilised by performing three washes with ethanol 70% under shaking for 5 min each. After sterilisation, due to PVDF hydrophobicity, ethanol was gradually replaced by sterile deionised water and microspheres were incubated in a 20% (*v*/*v*) FBS aqueous solution overnight. Then, microspheres were washed three times with deionised water and resuspended in a solution of HRP/Gel at a volume ratio of 10/80 (mL of HRP/mL of Gel).

Basal medium and standard temperature, humidity and CO_2_ concentration conditions were used to expand MSCs [31]. Cell passages above 5 were not reached for any of the experiments presented here.

Cells were resuspended at 1 × 10^6^ cells/mL in the HRP/Gel solution containing the microspheres. 45 µL of cell suspension were cross-linked by adding 5 µL of H_2_O_2_ on each well of a 48-well plate and left in an incubator for 15 min to ensure hydrogel cross-linking. Once crosslinked, a cell culture medium was added.

#### 4.4.2. Cell Viability Assessment

After 24 h, the viability of MSCs encapsulated with PVDF, and PVDF-CFO microspheres were evaluated. Hoechst 3342 (1.5 µg/mL, Thermo Fisher, Waltham, MA, USA) and propidium iodide (1.5 µg/mL, Sigma-Aldrich) were added to the cell culture medium and incubated for 20 min at 37 °C. After incubation with fluorescent probes, cells were imaged using the INCELL 6000 Analyzer system (GE Healthcare, Chicago, IL, USA). Four randomised visual fields per well were analysed, and viability was determined using ImageJ software and applying Equation (4):(4)$$Viability\;\left(\%\right)=\frac{blue\;counts-red\;counts}{blue\;counts}\;x\;100$$

Gelatin hydrogels without microspheres (Gel) were used as viability controls.

#### 4.4.3. Cell Spreading and Microsphere Distribution

Cell spreading and microsphere distribution within the gelatin matrix were evaluated after 1 and 14 days of culture. Hydrogels were fixed in paraformaldehyde 4% (*v*/*v*) for 15 min at room temperature. Subsequently, samples were submerged in sucrose (Sigma-Aldrich) solution 30% (*w*/*v*) overnight, immersed in OCT (Tissue Tek) and stored at −80 °C. Hydrogel sections of 30 μm were obtained using a cryostat (Leica CM 1860 UV) and placed on SuperFrost slides (Thermo Scientific).

Samples were washed twice with Dulbecco’s Phosphate Buffer Saline ((DPBS) Gibco) and permeabilised using Triton X-100 (Sigma-Aldrich) 0.1% (*v*/*v*) in DPBS for 10 min. Permeabilisation solution was removed, and samples were washed twice with DPBS. Slides were incubated with Rhodamine Phalloidin (ActinRed 555 ReadyProbes Reagent, Invitrogen, Waltham, MA, USA), following the manufacturer’s instructions, and Hoechst 3342 (1:250) for 1 h. Slides were finally washed with DPBS, and a mounting medium was added. Representative images were taken using a fluorescence microscope (Nikon Eclipse 80i). Gelatin hydrogels without microspheres (Gel) were used as controls.

Non-cryosectioned hydrogels were also observed in a confocal microscope (Leica DMI8), following the same staining protocol, and image processing for 3D reconstructions was performed using ImageJ software.

#### 4.4.4. Piezoelectric Stimulation Influence on MSCs Proliferation

The influence of piezoelectric stimulation on MSCs proliferation was assessed by analysing cell metabolic activity on days 2, 7, 14 and 21 under static (no applied stimuli) and dynamic (cell culture under magnetic stimulation) conditions. An alternating magnetic field (0–230 Oe) was provided using a homemade magnetic bioreactor placed inside the incubator, applying a 0.3 Hz frequency and a 10 mm magnet displacement under the 48-well plate [61]. The stimulation program was divided into an active period of 16 h based on 5 min of magnetic stimulation and 25 min of resting time, followed by a non-active period of 8 h when no magnetic stimulation was applied [22,62]. A diagram of the magnetic stimulation program can be found in Figure 1.

At different time points, hydrogels were transferred to a new culture plate. The basal medium was replaced for DMEM without phenol red (Sigma-Aldrich) containing the tetrazolium salt MTS (3-(4,5-dimethylthiazol-2-yl)-5-(3-carboxymethoxyphenyl)-2-(4-sulfophenyl)-2H-tetrazolium) (Biovision) at a working dilution of 1:10. Hydrogels were incubated for 1 h at 37 °C. After that, the supernatant was transferred to a new plate and absorbance at 490 nm was read with a Victor3 microplate reader (PerkinElmer). Gelatin hydrogels without microspheres stimulated (S) and non-stimulated (NS) were used as controls.

#### 4.4.5. Influence of Piezoelectric Stimulation on MSCs Osteogenic Differentiation

Gene Expression Analysis

Gene expression of characteristic osteogenic markers was analysed to determine the influence of piezoelectric stimulation on MSCs’ osteogenic differentiation.

Gel-PVDF and Gel-PVDF-CFO hydrogels were seeded, and after 24 h of culture in the basal medium, it was replaced by a commercial osteogenic differentiation medium (PromoCell). Stimulated samples were placed in the bioreactor. After 7 and 14 days of culture, hydrogels were digested with collagenase 993 U/mL (Collagenase from Clostridium Histolyticum, Sigma-Aldrich) in DPBS for 30 min at 37 °C. Qiazol lysis reagent (Qiagen) and chloroform (Scharlab) were added with a ratio of 5:1 to purify nucleic acids. RNA was purified using an RNA extraction kit (RNeasy Micro Kit, Qiagen, Hilden, Germany), and the obtained concentration was measured by spectrophotometer (Nanodrop ONE, Thermo Scientific). 300 ng of total RNA were reverse transcribed using the Superscript III reverse transcriptase (Invitrogen) and oligo dT primers (Invitrogen), following the manufacturer’s instructions. Real-time qPCR was performed using LightCycler 480 SYBR Green I Master (Roche, Basel, Switzerland) in a LightCycler 480 Instrument (Roche), and amplifications were performed for 40 cycles. Primers used for amplification were designed from sequences found in the GeneBank database and are listed in Appendix A. For normalisation, glyceraldehyde-3-phosphate dehydrogenase (GAPDH) was used. 

Primer sequences were validated by dissociation curve/melt curve analysis. The relative changes in gene expression were calculated by E-method, applying Equation (5) [63]:Fold difference = (E_target_) ^Ct(target) calibrator − Ct(target) sample^/(E_normalizer_) ^Ct(normalizer) calibrator − Ct(normalizer) sample^(5)
where E is the efficiency of the target gene or the normaliser housekeeping gene GAPDH. Non-stimulated Gel-PVDF-CFO hydrogels were used as calibrators. The raw data were transferred using the LC480 conversion software (Version 2014), and then PCR efficiency for each pair of primers was calculated by LineReg PCR (Version 2021.1) [64].

Alkaline Phosphatase Activity Determination

Following the same cell culture protocol described in the previous section hydrogels were kept in culture for 7 and 14 days, and alkaline phosphatase (ALP) activity was assessed using SensoLyte ^®^ pNPP Alkaline Phosphatase Assay Kit (Anaspec, Fremont, CA, USA). Briefly, hydrogels were digested by adding 200 µL of collagenase 993 U/mL in 1X Assay Buffer for 30 min at 37 °C. After digestion, 50 µL of Triton X-100 (Sigma-Aldrich) 1.2% (*v*/*v*) in 1X Assay Buffer was added, and samples were incubated for 10 min at 4 °C in an orbital shaker. After that, samples were centrifuged at 4275 rpm for 10 min at 4 °C, and the supernatant was used to determine ALP activity following the kit’s manufacturer’s instructions. Acellular hydrogels were used as blanks.

ALP activity was normalised against cell metabolic activity determined by MTS assay, following the protocol described in Section 4.4.4. Non-stimulated Gel-PVDF-CFO hydrogels were used as controls.

### 4.5. Statistical Analysis

Cell culture experiments were performed, at least, in triplicates and a minimum of two technical replicates were used per technique. All results were expressed as mean ± standard deviation. Statistical analysis was performed on GraphPad Prism 9 (USA). Non-parametric Kruskal-Wallis and Dunn’s multiple comparison test were applied, and a 95% confidence interval was set to accept significant inter-group differences (*p*-value < 0.05).

## Data Availability

The data presented in this study are openly available and can be found at Riunet repository of Universitat Politècnica de València at http://hdl.hadle.net/10251/188261.

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
