# Peer review of "Magnetically Activated Piezoelectric 3D Platform Based on Poly(Vinylidene) Fluoride Microspheres for Osteogenic Differentiation of Mesenchymal Stem Cells"

_gels, 2022, doi:10.3390/gels8100680_

Round 1

Reviewer 1 Report

The manuscript reports the preparation of a 3D cell culture platform based on the combination of an injectable gelatin hydrogel containing microspheres of poly(vinylidene) fluoride (PVDF) and cobalt ferrite oxide (CFO). PVDF-CFO microspheres provide electroactivity to the hydrogel. Hydrogels were tested in vitro to assess the cytocompatibility on mesenchymal stem cells, and following the piezoelectric stimulation, the activation of the osteogenic differentiation pathway was demonstrated. The manuscript falls within the aim and scope of Gels, it is well written, and the experiments were carefully done.  

 I only have the following minor comments that have to be addressed:

·         Regarding cytocompatibility, a significant difference was observed at 21 days when incorporating PVDF-CFO microspheres in the gel and following stimulation. This result should be better discussed.

·         Proper mineralization of the hydrogels, following MSC differentiation, should be demonstrated qualitatively by confocal microscopy and quantitatively by colourimetric assay (es. Alizarin red).

Reviewer 2 Report

Dear Author,

This is nice piece of work.

1.Size of microsphere is in um what size of biological cell effectively gown on it ?I did not see any literature control to comapare your results?

2.What is novelty (selling point)?

3. Poly(Vinylidene) Fluoride well studied in tissue engineering for nanofibers?

4.Which routes of administration you are using for animal or human testing?

5.Microsphre mostly given as oral or Parenteral dosage form so in your study i did not se porous nature so effectively allow cell growth?

6.If you are injecting this microsphere how it will clear from circulation?

7.Poly(Vinylidene) Fluoride is FDA approved for injectable?

8.Is there any regulatory concerned for Poly(Vinylidene) Fluoride?

Round 2

Reviewer 2 Report

Dear Author,

I am happy to read your comments and it was nicely improved for scientific community.Its good to change title as misleading and confusion.

Best luck

Regards

PSG